# Risk Prediction of Chronic Rhinosinusitis with or without Nasal Polyps in Taiwanese Population Using Polygenic Risk Score for Nasal Polyps

**DOI:** 10.3390/biomedicines11102729

**Published:** 2023-10-09

**Authors:** Rong-San Jiang, I-Chieh Chen, Yi-Ming Chen, Tzu-Hung Hsiao, Yi-Chen Chen

**Affiliations:** 1Departments of Medical Research, Taichung Veterans General Hospital, Taichung 407219, Taiwan; rsjtaiwan@yahoo.com.tw (R.-S.J.); icchen@vghtc.gov.tw (I.-C.C.); ymchen1@vghtc.gov.tw (Y.-M.C.); thsiao@vghtc.gov.tw (T.-H.H.); 2Departments of Otolaryngology, Taichung Veterans General Hospital, Taichung 407219, Taiwan; 3School of Medicine, Chung Shan Medical University, Taichung 402306, Taiwan; 4RongHsing Research Center for Translational Medicine, National Chung Hsing University, Taichung 402202, Taiwan; 5Department of Post-Baccalaureate Medicine, College of Medicine, National Chung Hsing University, Taichung 402202, Taiwan; 6Division of Allergy, Immunology and Rheumatology, Taichung Veterans General Hospital, Taichung 407219, Taiwan; 7School of Medicine, National Yang-Ming Chiao Tung University, Taipei 112304, Taiwan; 8Institute of Biomedical Science, National Chung Hsing University, Taichung 402202, Taiwan; 9Precision Medicine Research Center, College of Medicine, National Chung Hsing University, Taichung 402202, Taiwan; 10Department of Public Health, Fu Jen Catholic University, New Taipei City 242062, Taiwan; 11Institute of Genomics and Bioinformatics, National Chung Hsing University, Taichung 402202, Taiwan

**Keywords:** chronic rhinosinusitis, nasal polyps, polygenic risk score

## Abstract

The association between single nucleotide polymorphisms and chronic rhinosinusitis (CRS) has been determined. However, it was not known whether the polygenic risk score (PRS) for nasal polyps (NP) could predict CRS with NP (CRSwNP) or without NP (CRSsNP). The aim of this study was to investigate the association between PRSs for NP and the risk of CRS with or without NP. Data from 535 individuals with CRS and 5350 control subjects in the Taiwan Precision Medicine Initiative project were collected. Four PRSs for NP, including PGS000933, PGS000934, PGS001848, and PGS002060 from UK Biobank, were tested in these participants. They were divided into four groups according to quartiles of PRSs. The logistic regression model was performed to evaluate CRSwNP and CRSsNP risk according to PRSs for NP. The PGS002060 had the highest area under the curve at 0.534 for CRSsNP prediction and at 0.588 for CRSwNP prediction. Compared to subjects in the lowest PRS category, the PGS002060 significantly increased the odds for CRSsNP by 1.48 at the highest quintile (*p* = 0.003) and by 2.32 at the highest quintile for CRSwNP (*p* = 0.002). In addition, the odds for CRSwNP increased by 3.01 times in female CRSwNP patients (*p* = 0.009) at the highest quintile compared with those in the lowest PRS category. The PRSs for NP developed from European populations could be applied to the Taiwanese population to predict CRS risk, especially for female CRSwNP.

## 1. Introduction

The pathogenesis of chronic rhinosinusitis (CRS) is not yet well understood [1]. Recent studies have suggested that genetic predisposition appears to increase the risk of developing CRS [2,3]. It has been believed that the genetics of CRS contribute to the study of the etiology of CRS [4,5]. Although monogenic or polygenic variations have been identified in the majority of individuals with CRS [6], the incremental effects induced by the combination of variations in multiple genes for CRS have not been conducted [7].

CRS is phenotypically classified into CRS with nasal polyps (CRSwNP) or without nasal polyps (CRSsNP) based on the absence or presence of nasal polyps (NP) [8]. The CRSwNP tends to have severe symptoms and a poorer quality of life and is more recalcitrant and difficult to treat than the CRSsNP [9]. Therefore, early diagnosis of CRSwNP can prevent patients from severe symptoms and complications, improve the quality of life for patients, and reduce healthcare costs [10]. CRSwNP has been diagnosed using several methods, including nasal endoscopy, CT scans, and some biomarkers, such as eosinophils, IL-5, and T_H_2 cytokines [8,11]. However, these methods cannot predict individuals at high risk of developing CRSwNP.

Previous studies have identified a lot of genes associated with CRSwNP, including *TLR2*, *TLR4*, *TAS2R38*, *NOS2*, *KRT19*, and *Periostin* [12,13,14,15,16]. Moreover, several genes, such as *IL18R1*, *CYP2S1*, *CCL26*, *POSTN*, *CST1*, and *TGFB1*, have been recognized to be related to NP [17,18,19,20,21]. Only two genes, *ALOX15* and *HLA-DQA1*, have been reported to be significantly associated with both NP and CRS in genome-wide association study (GWAS) meta-analysis [19]. However, no polygenic risk scores (PRS), which were calculated based on variations in multiple genes, have been established for CRSwNP.

Polygenic risk score (PRS), which summarizes an individual’s genetic component for a particular trait or disease by aggregating information from multiple genetic variants into a single score, is an approach to estimating genetic predisposition on disease liability [22]. Currently, some PRSs for NP have been revealed by GWAS of European ancestry UK Biobank, such as PGS000933, PGS000934, PGS001848, and PGS002060, which consisted of 433, 129, 1783, and 553,424 single nucleotide polymorphisms (SNPs), respectively [22,23]. However, whether these PRSs for NP could be a new potential method predicting the risk for patients developing CRSsNP or CRSwNP has not been determined yet. Moreover, only a very small proportion of participants in the UK Biobank were of Asian descent. The aim of our study is to evaluate the impact of PRSs for NPs on individuals with CRS in a Taiwanese population.

## 2. Materials and Methods

### 2.1. Study Population

All participants involved in this case-control study were aged 20 years or older and were enrolled in the Taiwan Precision Medicine Initiative (TPMI) project, whose medical records and blood were collected from Taiwanese volunteers at Taichung Veterans General Hospital (TCVGH) between June 2019 and May 2021. The diagnosis of CRS was made according to the EPOS criteria based on patient history, nasal endoscopy, and a CT of the sinuses [24]. Individuals with CRS were phenotypically divided into both CRSwNP and CRSsNP based on the presence or absence of nasal polyps, as seen under a nasal endoscopy. The subjects without CRS and other upper airway disorders, who were identified using International Classification of Diseases, Ninth Revision (ICD-9) diagnosis codes, including acute respiratory infections (ICD-9 code 460, 461.x, 462, 463, 464.x, 465.x, 466.x) and other diseases of upper respiratory tract (ICD-9 code 470, 472.x, 474.x, 476.x, 477.x, 478.x) were selected as controls [25]. Written consent for genetic analysis was received from all enrolled participants at the time of blood collection. This study was approved by the ethics committee of the TCVGH Institutional Review Board (IRB No. CE23130B).

### 2.2. Genotyping

To genotype the DNA extracted from blood specimens, we used the Axiom Genome-Wide TWB 2.0 Array Plate (Affymetrix, Santa Clara, CA, USA), which was designed specifically for Taiwan’s Han Chinese population and contained 714,431 SNPs [26]. The DNA was eliminated from the dataset if it had a missing rate exceeding 0.02, an inbreeding coefficient greater than 0.15, or a sex mismatch. Furthermore, we utilized Affymetrix Power Tools software (Version 1.20.0) to ensure data quality and excluded SNPs that failed the Hardy–Weinberg equilibrium tests with a *p*-value of less than 1.0 × 10^−5^, had a minor allele frequency of less than 0.05, or had a genotype missing rate greater than 5%. A total of 591,048 SNPs were confirmed to be appropriate for the GWAS following quality control. The SNPs were further aligned to the human genome reference Genome Reference Consortium Human Build 38 (GRCh38), and genotype imputation was carried out using the Michigan Imputation Server and the “minimac4” algorithm [27]. The 1000 Genomes Phase 3 (Version 5) reference panel was utilized during the imputation process [28]. The resulting dataset consisted of 4.9 million sites and 2504 samples after quality control. Approximately 3 million additional genetic variants were imputed after excluding variants that did not meet quality control criteria. The final report includes all biallelic variants that met the INFO score threshold of ≥0.3. The genotype data of each participant were merged with the selected SNPs using imputation data as a reference.

### 2.3. Polygenic Risk Scoreconstruction

To compute the PRS, we used the function named score provided by PLINK version 2.0 to aggregate the effects of risk alleles for each variant weighted by the size of their effect in the GWAS [29]. PLINK 2.0 can automatically resolve the issue of inverted effect and non-effect alleles between datasets. To avoid multicollinearity issues in statistical modeling during PRS development, linkage disequilibrium (LD) pruning and clumping were performed using PLINK2.0 to select independent and informative SNPs [30,31]. LD is the non-random association of alleles at different genetic loci in a population, which can result from physical proximity on a chromosome or other evolutionary factors. Two genetic variants, such as SNPs, that exhibit strong LD are more frequently inherited together than by chance. The PRSs used in this study were derived from the United Kingdom (UK) Biobank, including PGS000933, PGS000934, PGS001848, and PGS002060. The PGS000933 and PGS000934 comprised 433 and 129 SNPs, respectively, that were associated with NPs identified solely from individuals of European ancestry [23]. On the other hand, PGS001848 and PGS002060 consisted of 1783 and 553,424 SNPs, respectively, that were correlated with NPs confirmed by a Trans-ancestry GWAS meta-analyses [22]. The SNPs and their corresponding effect sizes used to construct these PRSs were obtained from the polygenic score catalog [32]. The approach we used was to weigh each allele dosage by its effect size, as previously described [33]. The standard equation used to calculate a weighted polygenic risk score is as follows:$${PRS}_{j}=\sum_{i}^{N}\beta_{i}\times {dosage}_{ij}$$

*N* represents the total number of SNPs in the polygenic score.

$\beta_{i}$ represents the effect size (often denoted as beta) associated with variant *i*.

${dosage}_{ij}$ stands for the number of copies of SNP*i* present in the genotype of individual *j*.

### 2.4. Statistical Analysis

The differences in continuous variables such as age and PRS score in demographic data presented as mean ± standard deviation (SD) were compared via Student’s *t*-test. Furthermore, the gender was shown as a number (percent) and analyzed using the Chi-square test. The predictive performance of various PRS for patients with CRS and CRSwNP was determined by receiver operating characteristic (ROC) curve analysis. All participants were further assigned into four groups according to PRS quartiles, which provided efficiently valuable effect size for risk evaluation of CRS and CRSwNP [33]. Each group contained 25% of the total participants. The first group was denoted as Q1, which spanned between the minimum value and the first quartile of PRS (0–25%). The second group was denoted as Q2, which spanned between the lower quartile and medium of PRS (26–50%). The third group was denoted as Q3, which spanned between the medium and the upper quartile of PRS (51–75%). The fourth group was denoted as Q4, which spanned between the upper quartile and the maximum value of PRS (76–100%). The association of the PRS status (the highest quartile versus the lowest quartile) and the status of CRS or CRSwNP were determined by univariate logistic regression analysis to adjust potential confounders. All statistical analyses were performed with IBM SPSS statistical software for Windows (Version 22.0, IBM Corp., Armonk, NY, USA).

## 3. Results

### 3.1. Baseline Characteristics and Polygenic Risk Score

A total of 535 individuals with CRS, including 172 CRSwNP and 363 CRSsNP ones, along with 5350 control subjects who met final inclusion criteria and were matched for age and sex with the individuals with CRS at a ratio of 1:10 were enrolled in this study. Descriptive characteristics that included age, sex, bilateral regions of NP, and PRS scores of 5885 participants of the entire cohort are shown in Table 1. The average scores of PGS000933, PGS000934, PGS001848, and PGS002060 were significantly higher in patients with CRS than in control individuals (*p* = 0.009, 0.015, 0.006, and 0.008, respectively). However, remarkably higher average scores of PGS001848 and PGS002060 were demonstrated in patients with CRSwNP than in patients with CRSsNP (*p* = 0.003 and 0.001, respectively).

### 3.2. Predictive Performance of PRS for Individuals with CRS or CRSwNP

The areas under curves (AUCs) of PGS000933, PGS000934, PGS001848, and PGS002060 were 0.533 [95% confidence interval (CI): 0.507–0.560], 0.531 (95% CI: 0.505–0.556), 0.533 (95% CI: 0.507–0.560), and 0.534 (95% CI: 0.508–0.560) for the discrimination between individuals with CRS and control subjects. Moreover, the AUCs of PGS000933, PGS000934, PGS001848, and PGS002060 were 0.532 (95% CI: 0.481–0.583), 0.517 (95% CI: 0.465–0.569), 0.580 (95% CI: 0.527–0.632), and 0.588 (95% CI: 0.537–0.640) for the differentiation between CRSwNP patients and CRSsNP ones.

All participants were further categorized into four groups (Q1 to Q4) based on the quartile of PGS001848 and PGS002060 scores to evaluate the risk for CRS or CRSwNP. The events of CRS and CRSwNP gradually decreased as the quartiles of PGS002060 increased, but they did not decrease in order as the quartiles of PGS001848 increased. The CRS event rates for all participants subgrouped into Q1 to Q4 of PGS001848 and PGS002060 were 11.0%, 9.2%, 7.8%, and 8.4%, as well as 10.5%, 9.2%, 9.4%, and 7.3%, respectively (*p* = 0.015 and 0.006, respectively, Figure 1A,C). The risk of CRS was significantly higher in Q1 compared to Q4 in PGS001848 and PGS002060 [odds ratio (OR) = 1.352 and 1.477, 95% CI: 1.058–1.729 and 1.142–1.910, *p* = 0.016 and 0.003, respectively]. Similarly, the CRSwNP event rates for individuals with CRS assigned into Q1 to Q4 of PGS001848 and PGS002060 were 41.4%, 37.3%, 23.1%, and 26.9%, as well as 42.1%, 33.6%, 29.1%, and 23.9%, respectively (*p* = 0.004 and 0.001, respectively, Figure 1B,D). The risk of CRSwNP was noticeably higher in Q1 compared to Q4 PGS001848 and PGS002060 (OR = 1.920 and 2.318, 95% CI: 1.147–3.212 and 1.371–3.921, *p* = 0.013 and 0.002, respectively). The PGS002060 yielded the best predictive performance because it showed the highest AUC and OR when comparing individuals with CRS with control subjects, as well as when comparing CRSwNP patients with CRSsNP ones.

### 3.3. Risk Stratification

Notably, the CRSwNP events in male CRS patients (36.8%, 106/288) were significantly higher than those in female patients with CRS (26.7%, 66/247) (*p* = 0.013), as shown in Table 1. To evaluate the potential confounding effect due to gender, the individuals with CRS were stratified into two groups by gender, and they were further categorized into four groups (Q1 to Q4) according to the quartile of PGS002060 score to evaluate the risk for CRSwNP. It is interesting to note that among female individuals with CRS, the descending order of CRSwNP events was as follows: Q1, Q2, and Q3 were equal to Q4 (Figure 2A). In contrast, among male patients, the descending order was Q1, Q3, Q2, and Q4 (Figure 2B). The highest risk of CRSwNP events was observed in female individuals with CRS in Q1 compared to Q4, with an OR of 3.007 (95% CI: 1.312–6.895, *p* = 0.009). Similarly, male individuals with CRS in Q1 had a higher risk of CRSwNP events compared to Q4, with an OR of 2.055 (95% CI: 1.033–4.088, *p* = 0.040).

## 4. Discussion

The present study aimed to investigate the association between PRS for NP and CRS, as well as its subtype CRSwNP, in a Taiwanese population. The results showed that individuals with CRS had significantly higher average scores of PGS000933, PGS000934, PGS001848, and PGS002060 compared to control individuals. Moreover, patients with CRSwNP had remarkably higher average scores of PGS001848 and PGS002060 than patients with CRSsNP. The predictive performance of PRS for differentiating between patients with CRS and control individuals, as well as for differentiating between patients with CRSwNP and CRSsNP, was also evaluated. The results showed that PGS002060 yielded the best predictive performance for both comparisons.

An association between genetic variants and NP along with CRS has been reported by the GWAS meta-analysis in a recent study [19]. Several genetic loci associated with CRSwNP were identified, including genes involved in immune response and inflammation [14,15,16]. These studies support the role of genetic factors in the development of CRS and CRSwNP. However, the present study is the first to report an association between PRS and CRSwNP in a Taiwanese population. The significantly higher average scores of PGS001848 and PGS002060 in patients with CRSwNP suggest that these PRSs may be specifically associated with the development of NP in patients with CRS.

The findings of this study suggest that PRS based on genetic variants associated with immune response and inflammation may be useful in discriminating individuals with CRS from control subjects, as well as differentiating between CRSwNP and CRSsNP patients. The PGS002060 score showed the highest predictive performance and may be a useful tool in identifying individuals at higher risk for CRSwNP. The risk for CRS and CRSwNP was further estimated based on quartiles of the PGS002060 score. The results showed that the events of both CRS and CRSwNP decreased as the quartiles of PRS increased (Figure 1C,D). This suggests that individuals with higher PRS scores may have a higher risk of developing CRS and CRSwNP. This highlights the importance of considering genetic factors in the diagnosis of CRS, particularly in CRSwNP patients. Furthermore, this study provides evidence for the potential use of PRS as a new method for predicting the personalized risk of developing CRS and CRSwNP based on an individual’s genetic information. However, further research is needed to validate these findings and determine how they can be translated into clinical practice.

Considering the higher prevalence of CRSwNP in males compared to females, it is important to take into account the potential confounding effect of gender on the association between PRS and CRSwNP [34,35]. The results of this study demonstrated that male patients with CRS had higher rates of CRSwNP events (36.8%, 106/288) in comparison to female patients with CRS (26.7%, 66/247). Furthermore, this study found that females with CRS in Q1 had a higher risk of CRSwNP events compared to Q4, with an odds ratio (OR) of 3.007 (Figure 2A), while male individuals with CRS in Q1 also had a higher risk of CRSwNP events than those in Q4, but with a lower OR of 2.055 (Figure 2B). This finding suggests that gender-specific genetic factors may have a different impact on the development of CRSwNP in males and females. This emphasizes the need for further research to investigate gender-specific genetic factors associated with CRSwNP.

In terms of the pathogenesis of CRS, it shares some similarities with asthma [36]. Specifically, they can both be differentiated into endotypes based on the type of inflammation present, with type 2 eosinophilic inflammation being a common feature [36]. This suggests that there may be some shared underlying mechanisms driving the inflammatory response in these conditions. Additionally, there are some similarities in the remodeling process that occurs in the upper and lower respiratory tracts in these conditions [37]. In fact, a previous study has shown that patients with severe asthma had a higher incidence of NPs compared to patients with mild/moderate asthma [38]. This suggests that the same SNPs associated with immune response and inflammation that were used to develop the PRS used in this study may also be relevant to asthma.

Furthermore, some of the PRS items, such as PGS001341, PGS001343, PGS001344, PGS001345, and PGS001346, have previously been associated with asthma risk in the same solely individuals of European ancestry for constructing PRSs used in this study [23]. Therefore, it is possible that these PRS items may also be useful in predicting asthma risk in individuals. However, it is important to note that these associations have only been observed in individuals of European ancestry, and further research would be needed to confirm this hypothesis and determine the extent to which the same PRS items are relevant to both CRS and asthma.

It should be noted that the present study has some limitations. Firstly, this study was conducted in a Taiwanese population, which may limit the generalizability of the findings to other populations. The transferability of genetic findings between different racial and ethnic groups can be limited. To improve the transferability of genetic findings, it is important to increase the diversity of study populations in the initial discovery phase of the PRS development [39]. This can help identify genetic variants that are relevant across different populations and reduce the risk of population-specific biases. The PRSs we used in the present study were obtained from the UK Biobank, which has a large and diverse sample size of over 500,000 individuals, including individuals of different ancestries such as European, African, South Asian, East Asian, and Hispanic/Latino. However, only a very small proportion of participants in the UK Biobank were of Asian descent, with East Asian participants making up only 9%.

Secondly, this study did not account for potential environmental factors that could contribute to the development of CRS or CRSwNP. Environmental factors such as air pollution, occupational exposure to certain chemicals, and exposure to tobacco smoke have been identified as potential risk factors for CRS [40]. Further research is needed to fully understand the complex interplay of environmental factors that contribute to the development of CRS.

Thirdly, this study used a PRS to predict CRSwNP risk without investigating specific genetic variants associated with CRSwNP. This limitation prevents further insight into the etiology and underlying genetic mechanisms of CRSwNP, which could potentially lead to the development of more targeted and effective treatments for CRSwNP.

## 5. Conclusions

The present study provides evidence for an association between PRS for NP and CRS and CRSwNP in a Taiwanese population. These findings suggest that genetic factors may play a role in the development of CRS and its subtypes, particularly in patients with CRSwNP. This study also highlights the importance of considering gender-specific genetic factors in the diagnosis and treatment of CRSwNP. Further studies are needed to validate these findings and identify specific genetic variants associated with CRSwNP. These findings suggest that there may be significant differences in the risk of CRSwNP events among different quintiles, particularly in female patients.

## Figures and Tables

**Figure 1 biomedicines-11-02729-f001:**
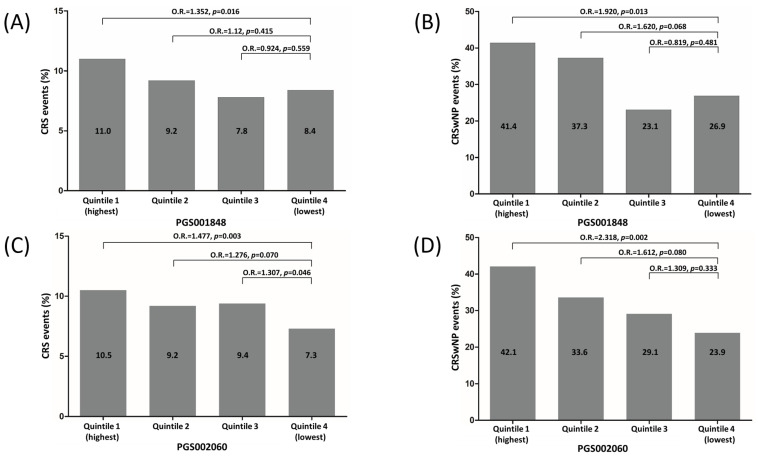
Percentage of CRS events in all participants (**A**,**C**), and CRSwNP events in individuals with CRS (**B**,**D**) by quintiles of PRS.

**Figure 2 biomedicines-11-02729-f002:**
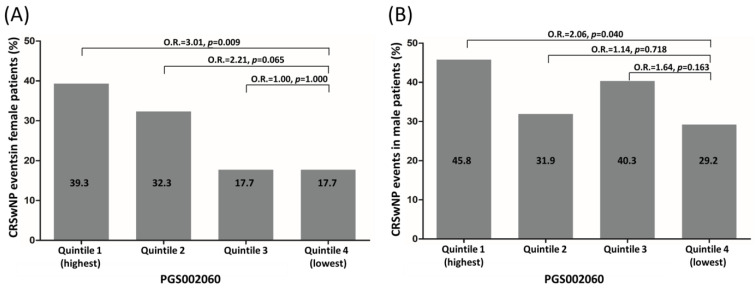
Frequency of CRSwNP events in female patients with CRS (**A**) and in male patients having CRS (**B**) by quintiles of PRS.

**Table 1 biomedicines-11-02729-t001:** Demographics of participants in this study.

Factors	Evaluation of CRS			Evaluation of Nasal Polyps		
	Patients with CRS N = 535	Control Individuals * N = 5350	*p*	Patients with CRSwNP N = 172	Patients with CRSsNP N = 363	*p*
Age (years), mean ± SD	58.67 ± 13.78	58.60 ± 13.69	0.918	58.05 ± 13.30	58.89 ± 13.93	0.515
Men, n (%)	288 (53.8)	2888 (54.0)	0.947	106 (61.6)	182 (50.1)	0.013
Women, n (%)	247 (46.2)	2462 (46.0)		66 (38.4)	181 (49.9)	
Nasal polyps						
Right				44		
Left				49		
Bilateral				79		
PGS000933, mean ± SD	−0.1600 ± 0.2152	−0.1856 ± 0.2038	0.009	−0.1484 ± 0.1971	−0.1655 ± 0.2233	0.372
PGS000934, mean ± SD	−0.2177 ± 0.1340	−0.2319 ± 0.1281	0.015	−0.2156 ± 0.1291	−0.2187 ± 0.1365	0.798
PGS001848, mean ± SD	0.4392 ± 0.4684	0.3841 ± 0.4364	0.006	0.5251 ± 0.4711	0.3985 ± 0.4711	0.003
PGS002060, mean ± SD	−0.0039 ± 0.0050	−0.0044 ± 0.0048	0.008	−0.0028 ± 0.0050	−0.0044 ± 0.0050	0.001

Abbreviation: CRS, chronic rhinosinusitis; CRSwNP, chronic rhinosinusitis with nasal polyps; CRSsNP, chronic rhinosinusitis without nasal polyps; *p* < 0.05 indicates statistical significance. * Control individuals: age and sex-matched 1:10.

## Data Availability

All the data supporting the findings of this study can be found within this article.

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
