# Peer review of "Risk Prediction of Chronic Rhinosinusitis with or without Nasal Polyps in Taiwanese Population Using Polygenic Risk Score for Nasal Polyps"

_biomedicines, 2023, doi:10.3390/biomedicines11102729_

Round 1
Reviewer 1 Report
I appreciate the opportunity to review the manuscript for publication in MDPI Biomedicines. I feel that the topics are updated and interesting, and the manuscript is grossly well organized. The authors assessed whether the polygenic risk score (PRS) for nasal polyps (NP) could predict CRS with (CRSwNP) or without NP (CRSsNP). The results would draw good attention to readers. I have a few comments as follows.
L142: “However, remarkably higher average scores of PGS001848 and PGS002060 were only demonstrated in patients with CRSwNP than in patients with CRSsNP (P=0.003 and 0.001, respectively).”
The authors had better describe the definition of classifying 2 different phenotypes more in detail. BA comorbidity, eosinophil densities, bilateral regions, ethmoid dominance etc. These classification might lead to another significant factor responsible for the CRSwNP events other than gender shown in Figure 2.
L156: “All participants were further categorized into four groups (Q1 to Q4) based on quartile of PGS001848 and PGS002060 score to evaluate the risk for CRS or CRSwNP.”
The authors had better illustrate the quartile demarcation more in detail.
L233: “Thirdly, the study did not investigate specific genetic variants associated with CRSwNP, which may provide further insight into the etiology of CRSwNP.”
Although written as above, it is better to add further information on some examples associated with CRSwNP in relation to the present PRS setting. The authors had better discuss possible relation to other related diseases, i.e. BA and AIA in using the Polygenic Risk Score items.
Author Response
Dear Editor,
Thanks very much for handling our manuscript.
We are thankful the opportunity to revise our manuscript to address the comments raised during the editorial review.
In this version, we have modified the manuscript accordingly to reviewers’ comments. Moreover, we extend the revised manuscript more than 4000 words during revisions. The changes have been highlighted in red in the revised manuscript.
-------------------------------------------------------------------------------------------------------
Reviewer 1
Comments and Suggestions for Authors
I appreciate the opportunity to review the manuscript for publication in MDPI Biomedicines. I feel that the topics are updated and interesting, and the manuscript is grossly well organized. The authors assessed whether the polygenic risk score (PRS) for nasal polyps (NP) could predict CRS with (CRSwNP) or without NP (CRSsNP). The results would draw good attention to readers. I have a few comments as follows.
L142: “However, remarkably higher average scores of PGS001848 and PGS002060 were only demonstrated in patients with CRSwNP than in patients with CRSsNP (P=0.003 and 0.001, respectively).”
The authors had better describe the definition of classifying 2 different phenotypes more in detail. BA comorbidity, eosinophil densities, bilateral regions, ethmoid dominance etc. These classification might lead to another significant factor responsible for the CRSwNP events other than gender shown in Figure 2.
Response: Thanks for the suggestion. The CRS patients were classified into CRS with nasal polyps or CRS without nasal polyps based on the findings when thy received endoscopic examination. When nasal endoscopy was performed to examine these patients, if polyps were found in either nasal cavity, they were classified into CRS with nasal polyps, and if no polyps were found in both nasal cavities, they were classified into CRS without nasal polyps.
The distribution of nasal polyps in nasal cavity in patients with CRSwNP was listed in Table 1. Out of 172 individuals with CRSwNP, 93 had unilateral NP, with 44 on the right side and 49 on the left side. Additionally, 79 individuals with CRSwNP had bilateral NP.
L156: “All participants were further categorized into four groups (Q1 to Q4) based on quartile of PGS001848 and PGS002060 score to evaluate the risk for CRS or CRSwNP.” The authors had better illustrate the quartile demarcation more in detail.
Response: Thanks for the suggestion. The quartile breaks down the data into quarters so that 25% of the measurements are less than the lower quartile, 50% are less than the median, and 75% are less than the upper quartile. There are three quartile values including a lower quartile, median, and upper quartile to divide the data set into four ranges, each containing 25% of the data points.
In the revised manuscript we illustrate the quartile demarcation more in detail in Materials and Methods section as following:
“Each group contains 25% of the total participants. The first group is denoted as Q1 which points between the minimum value and the first quartile of PGS (0-25%). The second group is denoted as Q2 which points between the lower quartile and medium of PGS (25-50%). The third group is denoted as Q3 which points between the medium and the upper quartile of PGS (51-75%). The fourth group is denoted as Q4 which points between the upper quartile and the maximum value of PGS (76-100%).”
L233: “Thirdly, the study did not investigate specific genetic variants associated with CRSwNP, which may provide further insight into the etiology of CRSwNP.”
Although written as above, it is better to add further information on some examples associated with CRSwNP in relation to the present PRS setting. The authors had better discuss possible relation to other related diseases, i.e. BA and AIA in using the Polygenic Risk Score items.
Response: Thanks for the suggestion. In the revised manuscript, we have added more information about possible relation between CRS and asthma by using the Polygenic Risk Score items in Discussion section as following:
“In terms of the pathogenesis of CRS, it shares some similarities with asthma [36]. Specifically, they can both be differentiated into endotypes based on the type of inflammation present, with type 2 eosinophilic inflammation being a common feature [36]. This suggests that there may be some shared underlying mechanisms driving the inflammatory response in these conditions. Additionally, there are some similarities in the remodeling process that occurs in the upper and lower respiratory tracts in these conditions [37]. In fact, a previous study has shown that patients with severe asthma had a higher incidence of NPs compared to patients with mild/moderate asthma [38]. This suggests that the same SNPs associated with immune response and inflammation that were used to develop the PRS used in this study may also be relevant to asthma.
Furthermore, some of the PRS items, such as PGS001341, PGS001343, PGS001344, PGS001345, and PGS001346, have previously been associated with asthma risk in the same solely individuals of European ancestry for constructing PRSs used in this study [23]. Therefore, it is possible that these PRS items may also be useful in predicting asthma risk in individuals. However, it is important to note that these associations have only been observed in individuals of European ancestry and further research would be needed to confirm this hypothesis and determine the extent to which the same PRS items are relevant to both CRS and asthma.”
Reviewer 2 Report
Dear authors,
After the review process, I have several comments: the paper are ok as a communication, not as an article, if you agree this changes; table 1 should be moved as supplementary file; Risk prediction for chronic rhinosinusitis is an important aspect of healthcare, and you should add responses to some important questions related to risk prediction:
-
- What are the known risk factors for chronic rhinosinusitis?
-
- Can genetic markers or family history be used to predict CRS risk?
-
- Are there specific environmental factors that increase the risk of CRS?
-
- How can we assess the role of comorbidities in CRS risk prediction?
-
-
- What biomarkers can be used for early detection and prediction of CRS?
-
-
- How does age impact CRS risk, and can age-specific models be developed?
-
-
- Can the development of CRS risk prediction tools lead to early intervention and improved outcomes?
-
- What ethical considerations should be taken into account when implementing risk prediction for CRS?
Author Response
Dear Editor,
Thanks very much for handling our manuscript.
We are thankful the opportunity to revise our manuscript to address the comments raised during the editorial review.
In this version, we have modified the manuscript accordingly to reviewers’ comments. Moreover, we extend the revised manuscript more than 4000 words during revisions. The changes have been highlighted in red in the revised manuscript.
-------------------------------------------------------------------------------------------------------
Reviewer 2
Comments and Suggestions for Authors
After the review process, I have several comments: the paper are ok as a communication, not as an article, if you agree this changes; table 1 should be moved as supplementary file; Risk prediction for chronic rhinosinusitis is an important aspect of healthcare, and you should add responses to some important questions related to risk prediction:
Response: We appreciate the comments from reviewer. We have extended the revised manuscript more than 4000 words during revisions as an article. We also have added some of the responses into the revised manuscript.
- - What are the known risk factors for chronic rhinosinusitis?
Response: Chronic rhinosinusitis (CRS) is a complex disease with a variety of risk factors. Several genetic, comorbid, demographic, and environmental factors have been identified that may increase the risk of developing CRS.
- - Can genetic markers or family history be used to predict CRS risk?
Response: Several candidate genes have been implicated in the pathogenesis of CRS, including genetic variations in antigen presentation, proinflammatory response, type 2 inflammation, innate immunity, tissue remodeling, cellular responses to stress, membrane receptors, and ion channels. However, the small sample sizes of these studies and the lack of replication of the significant variants in larger cohorts with different ethnic populations may affect the reliability of these findings. Henmyr et al (2014) aimed to replicate 53 SNPs reported to be associated with CRS by previous candidate gene studies in a White population of European origin; however, the investigators only found 7 variants in 7 genes that were significantly associated with CRS susceptibility. Of these, only some SNPs in PARS2, TGFB1, and NOS1 reached the significant threshold after multiple testing.
The heritability of CRS has not yet been fully understood. However, familial clustering has long been reported in CRS, suggesting a genetic component to the disease (Bohman et al., 2015). In a large population-based study, first- and second-degree relatives of CRS patients were found to have an elevated risk for CRS, with first-degree relatives of CRSwNP patients having a 4.1-fold elevated risk and first-degree relatives of CRSsNP patients having a 2.4-fold elevated risk (Oakley et al., 2015).
Therefore, large, replicated studies in tight cohorts across diverse ethnic and geographical populations are a pressing need in studying CRS genetics.
- - Are there specific environmental factors that increase the risk of CRS?
Response: Environmental factors have been identified as potential risk factors for CRS. Exposure to air pollution, particularly in areas with above-average pollution levels, has been associated with an increased risk of CRS (Min et al., 2015). Occupational exposure to certain chemicals, such as wood dust, has also been linked to an increased risk of CRS, particularly in individuals who were exposed to high levels of wood dust for more than 10 years. Other environmental factors that have been associated with an increased risk of CRS include exposure to tobacco smoke, both as a current smoker and as a former smoker (Tammemagi et al., 2010). However, the association between tobacco smoke exposure and CRS is not as strong as other risk factors. Additionally, exposure to other environmental factors, such as air conditioning, swimming pools, and pets, has been studied, but the evidence for their association with CRS is inconclusive (Bhattacharyya et al., 2009; Yu et al., 2012).
- - How can we assess the role of comorbidities in CRS risk prediction?
Response: To assess the role of comorbidities in CRS risk prediction, it is important to conduct comprehensive medical evaluations of patients with CRS. This may involve obtaining a detailed medical history, performing physical examinations, and conducting diagnostic tests to identify comorbidities that may be contributing to the patient's symptoms. In addition, large-scale population-based studies should be used to identify comorbidities that are associated with an increased risk of CRS. By analyzing data from these studies, we can identify risk factors that may help predict the likelihood of developing CRS in patients with certain comorbid conditions.
- - What biomarkers can be used for early detection and prediction of CRS?
Response: Several biomarkers have been studied for their potential use in the early detection and prediction of CRS. These include various cytokines, chemokines, and other inflammatory markers that are involved in the pathogenesis of CRS (Gayatri et al., 2020). The interleukin-5and eosinophil cationic protein have been found to be elevated in patients with CRSwNP, which suggests a role for type 2 inflammation in the pathogenesis of CRSwNP (Cho et al., 2017). Additionally, interleukin-8 and matrix metalloproteinase-9 have been found to be elevated in patients with CRSsNP, which suggests a role for neutrophilic inflammation in the pathogenesis of CRSsNP (Huang et al., 2019). Other biomarkers that have been studied for their potential use in the early detection and prediction of CRS include total immunoglobulin E (IgE), periostin, and tissue eosinophilia (Stevens et al., 2016). However, further research is needed to determine the clinical utility of these biomarkers in the diagnosis and management of CRS.
- - How does age impact CRS risk, and can age-specific models be developed?
Response: Age is a significant risk factor for chronic rhinosinusitis (CRS), as the prevalence of CRS increases with age (Vaitkus et al., 2021). Older individuals are more likely to have comorbidities and immune system changes that can contribute to the development of CRS. Additionally, older individuals may have a longer duration of exposure to environmental factors that can contribute to CRS, such as air pollution or occupational exposures. The present study is a case-control study in which the cases and controls are matched for age and sex to avoid potential confounding effect due to age. Age-specific models for predicting CRS risk can be developed to identify individuals at high risk, but further validation is needed to determine their clinical utility.
- - Can the development of CRS risk prediction tools lead to early intervention and improved outcomes?
Response: The findings of this study suggest that PRS based on genetic variants associated with immune response and inflammation may be useful in discriminating individuals with CRS from control subjects, as well as differentiating between CRSwNP and CRSsNP patients. Furthermore, this study provides evidence for the potential use of PRS as a new method for predicting the personalized risk of developing CRS and CRSwNP based on an individual's genetic information. However, further research is needed to validate these findings and determine how they can be translated into clinical practice.
- - What ethical considerations should be taken into account when implementing risk prediction for CRS?
Response: When implementing risk prediction for CRS, it is important to consider several ethical considerations. First, privacy is a key consideration when using patient data for risk prediction. Patients must be informed about how their data will be used, and their data must be kept confidential and secure. Second, informed consent is important to ensure that patients are fully aware of the risks and benefits of risk prediction, and that they provide their consent before their data is used. Third, bias must be avoided to ensure that the risk prediction algorithms do not perpetuate or exacerbate existing biases in healthcare. Fourth, stigmatization must be avoided so that patients are not discriminated against based on their risk score. Fifth, transparency is important so that patients can understand how their risk score was calculated. Sixth, fairness should be ensured in the development and implementation of risk prediction algorithms, taking into account factors such as race, ethnicity, and socioeconomic status.
Reviewer 3 Report
General Comments
The primary objective of this study was to investigate the association between polygenic risk scores (PRSs) for nasal polyps (NP) and the risk of chronic rhinosinusitis (CRS), with or without NP. The findings suggest that PRSs for NP developed from European populations can be applied to the Taiwanese population to predict CRS risk, particularly in female CRSwNP cases. However, it's essential to acknowledge the limitations of this study and consider further research to explore the clinical implications and potential applications of these findings in the diagnosis and management of CRS.
Particular Comments
1. Objective: It's important to clearly state the primary objective of the study in the abstract. In this case, it seems to be the investigation of the association between polygenic risk scores (PRSs) for nasal polyps (NP) and the risk of chronic rhinosinusitis (CRS) with or without NP. Ensure this is explicitly mentioned at the beginning of the abstract.
2. Data Source: You mention that data was collected from the Taiwan Precision Medicine Initiative project and PRSs were obtained from the UK Biobank. It might be helpful to briefly explain why data from these sources were chosen and any potential implications of using data from different populations.
3. Significance: Explain the clinical or practical significance of your findings. How might this information be used in the context of CRS diagnosis or treatment?
4. Limitations: Consider adding a brief statement about the limitations of the study. For example, if there are potential limitations related to the use of data from different populations, or if there were any limitations in the study design or data collection.
5. Grammar and Language: Ensure that the language and grammar are accurate and clear throughout the abstract. For instance, you could replace "CRS patients" with "individuals with CRS" for clarity.
Grammar and Language: Ensure that the language and grammar are accurate and clear throughout the abstract. For instance, you could replace "CRS patients" with "individuals with CRS" for clarity.
Author Response
Dear Editor,
Thanks very much for handling our manuscript.
We are thankful the opportunity to revise our manuscript to address the comments raised during the editorial review.
In this version, we have modified the manuscript accordingly to reviewers’ comments. Moreover, we extend the revised manuscript more than 4000 words during revisions. The changes have been highlighted in red in the revised manuscript.
-------------------------------------------------------------------------------------------------------
Reviewer 3
Comments and Suggestions for Authors
The primary objective of this study was to investigate the association between polygenic risk scores (PRSs) for nasal polyps (NP) and the risk of chronic rhinosinusitis (CRS), with or without NP. The findings suggest that PRSs for NP developed from European populations can be applied to the Taiwanese population to predict CRS risk, particularly in female CRSwNP cases. However, it's essential to acknowledge the limitations of this study and consider further research to explore the clinical implications and potential applications of these findings in the diagnosis and management of CRS.
Particular Comments
- Objective:It's important to clearly state the primary objective of the study in the abstract. In this case, it seems to be the investigation of the association between polygenic risk scores (PRSs) for nasal polyps (NP) and the risk of chronic rhinosinusitis (CRS) with or without NP. Ensure this is explicitly mentioned at the beginning of the abstract.
Response: Thanks for the suggestion. We have added the aim of the study in the abstract as following:
“The aim of this study was to investigate the association between PRSs for NP and the risk of CRS with or without NP.”
- Data Source: You mention that data was collected from the Taiwan Precision Medicine Initiative project and PRSs were obtained from the UK Biobank. It might be helpful to briefly explain why data from these sources were chosen and any potential implications of using data from different populations.
Response: Thanks for the suggestion. In the revised manuscript we have added more information about data sources we chose in Materials and Methods section 2.2 and 2.3 as following:
“The SNPs were further aligned to the human genome reference Genome Reference Consortium Human Build 38 (GRCh38), and genotype imputation was carried out using the Michigan Imputation Server and the 'minimac4' algorithm [27]. The 1000 Genomes Phase 3 (Version 5) reference panel was utilized during the imputation process [28]. The resulting dataset consisted of 4.9 million sites and 2,504 samples after quality control. Approximately 3 million additional genetic variants were imputed after excluding variants that did not meet quality control criteria. The final report includes all biallelic variants that met the INFO score threshold of ≥0.3.”
“PLINK 2.0 can automatically resolve the issue of inverted effect and non-effect alleles between datasets. To avoid multicollinearity issues in statistical modeling during PRS development, linkage disequilibrium (LD) pruning and clumping were performed using PLINK to select independent and informative SNPs [30, 31]. LD is the non-random association of alleles at different genetic loci in a population, which can result from physical proximity on a chromosome or other evolutionary factors. Two genetic variants, such as SNPs, that exhibit strong LD are more frequently inherited together than by chance.”
“The approach we used is to weight each allele dosage by its effect size, as previously described [34]. The standard equation used to calculate a weighted polygenic risk score is:
N represents the total number of SNPs in the polygenic score.
represents the effect size (often denoted as beta) associated with variant i.
stands for the number of copies of SNP i present in the genotype of individual j.”
Moreover, the use of data from different populations can have potential implications for the generalizability and accuracy of PRSs. The transferability of genetic findings between different racial and ethnic groups can be limited. We have added more description about limitations in Discussion section of the revised manuscript as following:
“The transferability of genetic findings between different racial and ethnic groups can be limited. To improve the transferability of genetic findings, it is important to increase the diversity of study populations in the initial discovery phase of PRS development [36]. This can help to identify genetic variants that are relevant across different populations and reduce the risk of population-specific biases. The PRSs we used in the present study were obtained from the UK Biobank which has a large and diverse sample size of over 500,000 individuals, including individuals of different ancestries such as European, African, South Asian, East Asian, and Hispanic/Latino. However, only a very small proportion of participants in the UK Biobank were of Asian descent, with East Asian participants making up only 9%.”
3.Significance: Explain the clinical or practical significance of your findings. How might this information be used in the context of CRS diagnosis or treatment?
Response: Thanks for the suggestion. We have added the clinical significance of our findings in Discussion section of the revised manuscript as following:
“The findings of this study suggest that PRS based on genetic variants associated with immune response and inflammation may be useful in discriminating individuals with CRS from control subjects, as well as differentiating between CRSwNP and CRSsNP patients. The PGS002060 score showed the highest predictive performance and may be a useful tool in identifying individuals at higher risk for CRSwNP. The risk for CRS and CRSwNP was further estimated based on quartiles of PGS002060 score. The results showed that the events of both CRS and CRSwNP decreased as the quartiles of PRS increased (Figure 1C and 1D). This suggests that individuals with higher PRS scores may have a higher risk of developing CRS and CRSwNP. This highlights the importance of considering genetic factors in the diagnosis of CRS, particularly in CRSwNP patients. Furthermore, this study provides evidence for the potential use of PRS as a new method for predicting the personalized risk of developing CRS and CRSwNP based on an individual's genetic information. However, further research is needed to validate these findings and determine how they can be translated into clinical practice.”
- Limitations:Consider adding a brief statement about the limitations of the study. For example, if there are potential limitations related to the use of data from different populations, or if there were any limitations in the study design or data collection.
Response: Thanks for the suggestion. In the revised manuscript we pointed out the limitation of this study in Discussion section as following:
“It should be noted that the present study has some limitations. Firstly, the study was conducted in a Taiwanese population, which may limit the generalizability of the findings to other populations. The transferability of genetic findings between different racial and ethnic groups can be limited. To improve the transferability of genetic findings, it is important to increase the diversity of study populations in the initial discovery phase of PRS development [36]. This can help to identify genetic variants that are relevant across different populations and reduce the risk of population-specific biases. The PRSs we used in the present study were obtained from the UK Biobank which has a large and diverse sample size of over 500,000 individuals, including individuals of different ancestries such as European, African, South Asian, East Asian, and Hispanic/Latino. However, only a very small proportion of participants in the UK Biobank were of Asian descent, with East Asian participants making up only 9%.
Secondly, the study did not account for potential environmental factors that could contribute to the development of CRS or CRSwNP. Environmental factors such as air pollution, occupational exposure to certain chemicals, and exposure to tobacco smoke have been identified as potential risk factors for CRS [40]. Further research is needed to fully understand the complex inter-play of environmental factors that contribute to the development of CRS.
Thirdly, the study used a PRS to predict CRSwNP risk without investigating specific genetic variants associated with CRSwNP. This limitation prevents further insight into the etiology and underlying genetic mechanisms of CRSwNP, which could potentially lead to the development of more targeted and effective treatments for CRSwNP.”
5.Grammar and Language: Ensure that the language and grammar are accurate and clear throughout the abstract. For instance, you could replace "CRS patients" with "individuals with CRS" for clarity.
Response: Thanks for the reminder. We have corrected accordingly.
Round 2
Reviewer 1 Report
I appreciate the opportunity to review again the manuscript for publication in MDPI biomedicines. I reckon that the manuscript has been revised and improved in accordance with the reviewers’ comments.
Reviewer 2 Report
No additional comments